# Intronic Polyadenylation in Acquired Cancer Drug Resistance Circumvented by Utilizing CRISPR/Cas9 with Homology-Directed Repair: The Tale of Human DNA Topoisomerase IIα

**DOI:** 10.3390/cancers14133148

**Published:** 2022-06-27

**Authors:** Terry S. Elton, Victor A. Hernandez, Jessika Carvajal-Moreno, Xinyi Wang, Deborah Ipinmoroti, Jack C. Yalowich

**Affiliations:** Division of Pharmaceutics and Pharmacology, College of Pharmacy, The Ohio State University, Columbus, OH 43210, USA; hernandez.521@osu.edu (V.A.H.); carvajalmoreno.1@osu.edu (J.C.-M.); wang.13576@osu.edu (X.W.); ipinmoroti.2@osu.edu (D.I.)

**Keywords:** topoisomerase IIα, RNA splicing, alternative polyadenylation, intronic polyadenylation, CRISPR/Cas9, gene editing, homology directed repair, drug resistance

## Abstract

**Simple Summary:**

DNA topoisomerase IIα (170 kDa, TOP2α/170) resolves nucleic acid topological entanglements by generating transient double-strand DNA breaks. TOP2α inhibitors/poisons stabilize TOP2α-DNA covalent complexes resulting in persistent DNA damage and are frequently utilized to treat a variety of cancers. Acquired resistance to these chemotherapeutic agents is often associated with decreased TOP2α/170 expression levels. Studies have demonstrated that a reduction in TOP2α/170 results from a type of alternative polyadenylation designated intronic polyadenylation (IPA). As a consequence of IPA, variant TOP2α mRNA transcripts have been characterized that have resulted in the translation of C-terminal truncated TOP2α isoforms with altered biological activities. In this paper, an example is discussed where circumvention of acquired TOP2α-mediated drug resistance was achieved by utilizing CRISPR/Cas9 specific gene editing of an exon/intron boundary through homology directed repair (HDR) to reduce TOP2α IPA. These results illustrate the therapeutic potential of CRISPR/Cas9/HDR to impact drug resistance associated with aberrant IPA.

**Abstract:**

Intronic polyadenylation (IPA) plays a critical role in malignant transformation, development, progression, and cancer chemoresistance by contributing to transcriptome/proteome alterations. DNA topoisomerase IIα (170 kDa, TOP2α/170) is an established clinical target for anticancer agents whose efficacy is compromised by drug resistance often associated with a reduction of nuclear TOP2α/170 levels. In leukemia cell lines with acquired resistance to TOP2α-targeted drugs and reduced TOP2α/170 expression, variant TOP2α mRNA transcripts have been reported due to IPA that resulted in the translation of C-terminal truncated isoforms with altered nuclear-cytoplasmic distribution or heterodimerization with wild-type TOP2α/170. This review provides an overview of the various mechanisms regulating pre-mRNA processing and alternative polyadenylation, as well as the utilization of CRISPR/Cas9 specific gene editing through homology directed repair (HDR) to decrease IPA when splice sites are intrinsically weak or potentially mutated. The specific case of TOP2α exon 19/intron 19 splice site editing is discussed in etoposide-resistant human leukemia K562 cells as a tractable strategy to circumvent acquired TOP2α-mediated drug resistance. This example supports the importance of aberrant IPA in acquired drug resistance to TOP2α-targeted drugs. In addition, these results demonstrate the therapeutic potential of CRISPR/Cas9/HDR to impact drug resistance associated with aberrant splicing/polyadenylation.

## 1. Introduction

The human DNA topoisomerase IIα (170 kDa, TOP2α/170) enzyme functions as a homodimer to generate TOP2α/170-DNA covalent cleavage complexes to produce transient double-strand DNA breaks [1,2,3,4]. This allows for DNA strand passage to resolve topological entanglements that occur during fundamental biologic processes, such as replication and chromosomal dysjunction at mitosis [1,2,3,4]. TOP2α/170 is highly expressed in rapidly proliferating cells and is necessary for cell survival [1,4,5]. Therefore, TOP2α interfacial inhibitors/poisons (e.g., etoposide, mitoxantrone, and anthracyclines), are efficacious in the treatment of a variety of leukemias, lymphomas, and solid tumors [6,7,8]. Cytotoxic antitumor activity induced by these agents results from formation and stabilization of TOP2α/170-DNA covalent cleavage complexes and resultant accumulation of DNA breaks [9,10,11]. Importantly, however, acquired chemoresistance to TOP2α inhibitors/poisons often limits the efficacy of these drugs [12,13,14,15,16].

Chemoresistance can result from a wide variety of molecular mechanisms, including abnormal cell cycling, altered drug metabolism, aberrant drug transport/trafficking processes, cell DNA damage/repair dysregulation, death evasion, and increased or modified drug targets [17,18]. In the case of TOP2α interfacial inhibitors/poisons, acquired resistance is most often associated with a decrease in the expression levels of TOP2α/170 and/or its altered subcellular localization since the cytotoxicity of these treatments depends upon the formation and accumulation of TOP2α/170-DNA cleavage complexes in the nucleus [13,14,15,16].

It is now apparent that a form of alternative polyadenylation (APA), designated as intronic polyadenylation (IPA), plays a major role in mediating resistance to TOP2α inhibitors/poisons in several human leukemia cell lines (HL-60, CEM, and K562) [16,19,20,21,22,23]. This review provides a general overview of alternative splicing and APA with emphasis given to IPA. Specific examples of IPA mediated chemoresistance with a focus on aberrant TOP2α IPA will also be discussed. Finally, this article examines CRISPR/Cas9 (clustered regularly interspaced short palindromic repeats/CRISPR-associated system 9) gene editing with homology directed repair (HDR) [24,25,26,27,28] as a therapeutic strategy to circumvent chemoresistance by modulation of IPA.

## 2. The Spliceosome

Eukaryotic pre-messenger RNA (pre-mRNA) undergoes extensive processing in a co-transcriptional manner before it can be translated. First, a 7-methylguanosine cap is added to the 5′ end (i.e., 5′ cap) of the growing pre-mRNA by a 5′-to-5′ phosphate linkage [29]. Next, the spliceosome is assembled onto the pre-mRNA in a stepwise fashion and is responsible for the removal of intronic sequences and ligation of exons [30,31]. The pre-mRNA is then processed at its 3′ end by endonucleolytic cleavage which terminates transcription, followed by the synthesis of a poly(A) (i.e., polyadenylation) tail on the cleaved transcript by poly(A) polymerase (PAP) reviewed in [32,33,34,35]. After maturation, mRNAs are then exported from the nucleus to the cytoplasm and translated into protein.

Briefly, the spliceosome is composed of five small nuclear ribonucleoproteins (snRNPs), U1, U2, U4, U5, and U6, all of which harbor a distinct small nuclear RNA (snRNA) and multiple auxiliary RNA-binding proteins reviewed in [36,37,38,39]. The most fundamental *cis* elements that direct the spliceosome to the pre-mRNA include the 5′ and 3′ splice sites (SS) and the branch point (BP) [36,37,38,39]. The 5′ SS is a nine-nucleotide (YAG//GURAGU; Y denotes the pyrimidine nucleotides C and U; R denotes purine nucleotides A and G) consensus sequence which spans the exon/intron boundary (Figure 1A) [36,37,38,39]. Spliceosome assembly is initiated when the U1 snRNP (i.e., comprised of eleven components: U1 snRNA [164 nucleotides (nt)], U1-70K/SNRNP70, U1A/SNRPA, U1C/SNRPC, and seven small nuclear ribonucleoproteins) recognizes the 5′ SS by RNA:RNA (9 nt) base-pairing between U1 snRNA’s 5′-end. This complementarity is the foundation of efficient U1 snRNP recruitment (Figure 1B, Complex E) [36,37,38,39].

The 3′ SS harbors a polypyrimidine tract and a highly conserved AG dinucleotide at the 3′ end of the intron (YYYYYYYYYNCAG//G; the letter Y denotes the pyrimidine nucleotides C and U; the letter N denotes any nucleotide: G, A, C, U) (Figure 1A) which recruits the U2AF heterodimer (i.e., U2AF1 and U2AF2) (Figure 1B, Complex E) [36,37,38,39]. The U2AF1 subunit binds to the conserved AG dinucleotide and the U2AF2 subunit preferentially binds to the polypyrimidine tract. The splicing factor 1 (SF1) then interacts with the BP (YNURAY) which is present 15–30 nucleotides upstream from the intron/exon boundary (Figure 1A,B, Complex E) [36,37,38,39]. Once U2AF and SF1 are bound to the 3′ SS/BP, the U2 snRNP is assembled to the spliceosome through base-pair interactions [36,37,38,39] between the U2 snRNA and the BP (Figure 1B, Complex A) [40]. The preassembled U4/U6/U5 tri-snRNP complex is then recruited and, with extensive conformational changes (Figure 1B, Complex B and Complex C), two transesterification catalytic reactions are facilitated to form a lariat and intron excision (Figure 1B, Catalytic Steps 1 and 2), which subsequently results in exon/exon ligation and correctly spliced mRNA [36,37,38,39].

## 3. Alternative Splicing

High-throughput sequencing-based methods have established that over 95% of human genes generate at least two alternative spliced mRNA isoforms, and many of these variants alter the protein-coding potential of the transcripts [41,42,43,44]. The prototypes of alternative splicing can include the following examples [36,37,38,39,41] (Figure 2): Constitutive Splicing: where every exon is included in the mature mRNA (Figure 2A); Exon skipping: where a specific exon is excluded from the mature mRNA (Figure 2B); mutually Exclusive Exons: where different exons are selected to generate distinct mature mRNAs but never coexist in the same transcript (Figure 2C); Alternative 5′ and 3′ Splice Sites: where maturation of a pre-mRNA leads to the inclusion/exclusion of a partial sequence of an intron or exon in the mature mRNA (Figure 2D,E); Intron Retention: where the entire intronic sequence is retained in the mature mRNA (Figure 2F). Intron-retaining mRNAs are susceptible to nuclear intron detention and/or degradation [45] or nonsense-mediated decay (NMD) in the cytoplasm [46]. Alternatively, some intron-retaining mRNAs exported to the cytoplasm can undergo translation, as reviewed in [44]. Regardless of the nature of alternative splicing, the resulting transcripts can encode protein isoforms with divergent structures and functions, or these mRNAs can be degraded which would result in decreased protein expression. Consequently, cell differentiation, lineage determination, and tissue/organ development can be impacted [47,48,49]. Importantly, aberrant alternative splicing can contribute to many characteristics of cancer progression, as reviewed in [39,50,51,52].

## 4. Cleavage and Polyadenylation Factors (CPAFs)

Almost all mammalian mRNAs are polyadenylated at their 3′ ends. Cleavage and polyadenylation are regulated by trans-acting factors binding to *cis* elements which can include the consensus AAUAAA hexamer and upstream U-rich and UGUA elements and downstream U-rich and GU-rich sequences, which together are defined as the poly(A) site (PAS) where cleavage occurs 10–30 nucleotides downstream (Figure 3A), as reviewed in [32,33,34,35]. Importantly, the strength of a given PAS is increased in a combinatorial manner by the cooperative assembly of four CPAF complexes and many other RNA-binding proteins which have been shown to modulate PAS recognition [32,33,34,35].

Pre-mRNA 3′ end cleavage and polyadenylation is initiated when the cleavage stimulation factor (CSTF) complex (i.e., comprised of three proteins denoted CSTF1/CSTF50, CSTF2/CSTF64 or its paralog CSTF2T/CSTF64τ, and CSTF3/CSTF77) binds with downstream U-rich and GU-rich sequences through direct interactions with the CSTF2 or CSTF2T (Figure 3B) [32,33,34,35,53]. The cleavage factor I (CFI) complex (i.e., comprised of two proteins denoted NUDT21 [nudix hydrolase 21 or cleavage and polyadenylation specific factor 5, CPSF5/CFI25] and either CPSF6/CFI68 or CPSF7/CFI59) subsequently binds to UGUA motifs located upstream of the AAUAAA hexamer sequence through interactions with both subunits of this complex (Figure 3C) [32,33,34,35,53].

The binding of these complexes helps to stabilize the cleavage and polyadenylation specificity factor (CPSF) complex binding to the PAS [54]. Specifically, the CPSF complex (i.e., composed of six proteins; CPSF1/CPSF160, CPSF2/CPSF100, CPSF3/CPSF73, CPSF4/CPSF30, WDR33 [WD repeat domain 33], and FIP1L1 [factor interacting with PAPOLA and CPSF1]), binds to the AAUAAA hexamer through direct interactions with WDR33 and CPSF4/CPSF30, which simultaneously and synergistically recognize this sequence (Figure 3D) [55,56]. The FIP1L1 subunit of the CPSF complex is thought to bind to U-rich sequences upstream from the AAUAAA hexamer sequence (Figure 3D) [57]. The CPSF3/CPSF73 subunit of the CPSF complex is an endonuclease and catalyzes the cleavage reaction ∼10–30 nucleotides downstream from the AAUAAA hexamer for 3′ end processing of pre-mRNAs (Figure 3A,D) [58].

The cleavage factor II (CFII) complex (i.e., comprised of two subunits; CLP1 [cleavage factor polyribonucleotide kinase subunit 1] and PCF11 [PCF11 cleavage and polyadenylation factor subunit]) also contributes to pre-mRNA cleavage (Figure 3D) [34,58,59]. Finally, the CPSF1/CPSF160 and FIP1L1 subunits of the CPSF complex directly recruit polyA polymerase (PAP) to the pre-mRNA 3′ end cleavage site to initiate polyadenylation (Figure 3E) [55].

## 5. Alternative Polyadenylation

Alternative polyadenylation (APA), which occurs in 60–70% of protein coding genes [34,35,60,61,62,63], is a biological process where multiple PASs are differentially utilized to produce distinct mRNA isoforms in a cell- or tissue-specific manner. Like alternative splicing, APA is increasingly recognized as a widespread mechanism used to control gene expression, cell proliferation, and senescence, and is thought to play a major role in human disease [34,35,60,61,62,63].

There are four well-defined types of APA. The most common form of APA can occur when multiple PASs are located within the 3′-untranslated region (3′-UTR) of mRNAs (i.e., proximal and distal to the translation stop codon) and is designated as Tandem 3′-UTR APA (Figure 4A) [34,35,60,61,62,63]. Although the mRNA coding potential remains unchanged, the 3′-UTR lengths of these transcripts differ significantly (Figure 4A). Importantly, these differences may regulate gene expression at the posttranscriptional level by altering the availability of both microRNA-binding sites and RNA-binding protein sites, which in turn could alter mRNA stability, translational efficiency, nuclear export, and mRNA localization [64,65].

In contrast to Tandem 3′-UTR APA, three additional forms of APA can occur when PASs are located upstream of the last exon (Figure 4A–C). Notably, these additional types of APA are directly linked with spliceosome function/alternative splicing [66,67] and may result in changes to both the protein-coding potential and the 3′-UTR of an mRNA transcript. For example, Skipped Terminal Exon APA (Figure 4B) can arise when alternative splicing changes the last exon resulting in the utilization of a new PAS [34,35,60,61,62,63]. Internal Exon APA (Figure 4C) can occur when an upstream exon contains a cryptic PAS resulting in mRNA transcripts which lack an in-frame stop codon and are subsequently degraded by the non-stop decay pathway [68] or non-stop protein degradation (since polyA-tracts generate C-terminal poly-lysine tags) [69]. Finally, Intronic APA (IPA) (Figure 4D) can result in composite terminal exons through the inhibition of the 5′ SS between the exon/intron boundary and the subsequent utilization of a cryptic PAS harbored within this intron [34,35,60,61,62,63]. IPA results in the extension of an internal exon coding sequence into the adjacent intron (Figure 4D). IPA has also been defined as an example of alternative last exon (ALE) splicing [70], or premature transcription termination (PTT) [63].

## 6. Intronic Polyadenylation (IPA)

One of the first published examples of APA (i.e., IPA) (Figure 4D) involved genes encoding the immunoglobulin M heavy chain (IgM) proteins [71,72,73]. Mature B cells were shown to utilize the distal PAS located in the 3′-UTR to produce a full-length mRNA transcript that encoded a plasma membrane bound form of IgM. In contrast, after B cell activation, the plasma cell utilized a cryptic PAS located in an intron upstream from the two exons that encode the IgM transmembrane domain. This phenomenon results in a shortened mRNA which encodes a secreted IgM antibody [71,72,73]. Subsequently, Davis et al. [74], identified 376 mouse genes, by bioinformatic analysis, that potentially use IPA for regulating membrane anchoring.

It is now estimated that 20% of human genes have at least one IPA event [34,66] which is usually associated with weak 5′ SS, large introns, and strong PASs located ~100–1000 bp downstream from the 5′ SS of an intron (Figure 4D) [66]. Recent RNA-seq data demonstrated that IPA is a widespread event in both normal and malignant tissues, in blood-derived immune cells and is differentially utilized during B-cell development [75]. Additionally, it has been established that tumor-suppressor genes can be inactivated by IPA in patients with chronic lymphocytic leukemia [76]. Finally, it has recently been shown that Cyclin-dependent kinase 12 (CDK12) regulates DNA repair gene expression by suppressing IPA [77].

Importantly, mRNA transcripts that result from IPA can have multiple outcomes (Figure 4D). For example, when IPA occurs near the transcription start site, these mRNA transcripts are rapidly degraded [63]. In contrast, more distal IPA events can result in noncoding RNAs (ncRNAs) which may serve as a scaffold for RNA binding proteins [76] or they may harbor small open reading frames which can encode biologically important micropeptides [78]. Alternatively, IPA mRNA isoforms can encode truncated proteins that lack the C-terminal domain(s) present in the full-length parental protein which may result in proteins with physiologically distinct properties (e.g., membrane bound versus soluble) [79,80], or truncated dominant negative functions [16,19,20,21,22,23,81,82,83], thereby diversifying the transcriptome/proteome by C-terminal domain loss. Regardless of the fate of prematurely terminated transcripts (e.g., stable or unstable, ncRNA, encoding micropeptides or truncated proteins), IPA will attenuate the expression of the corresponding full-length mRNA/protein.

IPA is utilized to diversify normal cellular processes and contributes to various disease states including cancer and cancer treatment resistance [19,20,21,22,23,34,35,60,61,62,63]. Therefore, it is important to understand the mechanisms by which IPA can be regulated. Although not fully understood, it is clear that changes in the expression levels of specific CPAFs [59,63,67,75,84,85] and U1 snRNP, with respect to U1 telescripting [86], can modulate IPA.

## 7. The Tale (Tail) of TOP2α IPA in Acquired Chemoresistance: Part 1

The human *TOP2α* gene has been mapped to chromosome 17q21-22 and comprises 35 exons and spans ~35 kb (Figure 5A) [87]. The full-length mRNA transcribed from this gene is 5695 nt (Figure 5B) and the open reading frame encodes a protein comprising 1531 amino acids (aa), with a calculated molecular weight of 174,386 Da (i.e., TOP2α/170) (Figure 5C) [88].

TOP2α/170 primarily resides and functions in the nucleus as a homodimer. Each monomer utilizes a tyrosine active site (i.e., Tyr 805) (Figure 5C) to generate TOP2α/170-DNA covalent cleavage complexes to produce transient double-strand DNA breaks to resolve topological DNA entanglements by passing intact DNA double strands through the formed breaks [1,2,3,4]. TOP2α/170 contains three subunit dimerization interfaces/gates (Figure 5C) which function to regulate enzymatic DNA cleavage and DNA strand passage via successive opening/closing of these gates [89,90,91].

Given that TOP2α/170 enzymatic activity is necessary for cell survival, this enzyme has been targeted by interfacial inhibitors/poisons (e.g., etoposide, mitoxantrone, doxorubicin, daunorubicin, and analogs) [9,10,11]. These drugs inhibit the reversal/religation of transient TOP2α/170-mediated DNA double strand breaks by interaction within the scissile break sites generated by TOP2α/170 on the top and bottom strands of DNA [9,10,11]. Therefore, the stabilization of TOP2α/170-DNA covalent cleavage complexes leads to the accumulation of DNA breaks and ultimately cell death [9,10,11].

Although chemoresistance can result from a variety of molecular mechanisms [17,18], acquired resistance to TOP2α-targeted drugs has been shown to be associated with TOP2α pre-mRNA IPA [16,19,20,21,22,23]. For example, in mitoxantrone-resistant human acute myeloid leukemia HL-60 cells, generated by stepwise drug exposure, the resulting clonal cell line, HL-60/MX2, was 35-fold resistant to mitoxantrone and cross-resistant to a number of additional TOP2-targeting agents [92]. When compared to parental HL-60 cells, TOP2α/170 protein levels were reduced in HL-60/MX2 cells and a novel C-terminal truncated TOP2α isoform migrating at ~160 kDa (TOP2α/160) was detected in the cytoplasm [93]. These cells also expressed a unique ~4.8 kilobase (kb) TOP2α mRNA transcript [93].

Further characterization of the HL-60/MX2 cells established that the truncated TOP2α/160 isoform (1436 aa and a calculated molecular weight of 164,052 Da) was the translation product of a TOP2α mRNA (4550 nt) that was generated from a cryptic PAS harbored in intron 33 (i.e., I33 IPA) (Figure 6A–C) [19]. As a result of TOP2α I33 IPA, the TOP2α/160 isoform lacks 108 aa from the C-terminal domain of TOP2α/170. In addition, a unique 14 aa sequence is present, encoded by translation of the exon 33/intron 33 “read-through” (Figure 6C) [19]. The well-characterized nuclear localization signal (NLS), NLS 1454-1497 (Figure 6C) [94,95], was absent from the truncated TOP2α/160 isoform. These results suggested that the aberrant nuclear-cytoplasmic localization of the truncated TOP2α/160 isoform, as a result of IPA, and the subsequent decrease of the wildtype TOP2α/170 expression levels, play a role in mediating mitoxantrone resistance in HL-60/MX2 cells [19].

## 8. The Tale (Tail) of TOP2α IPA in Acquired Chemoresistance: Part 2

Our laboratory generated a TOP2α interfacial inhibitor/poison-resistant cell line to investigate the molecular mechanisms by which acquired resistance can arise [94,95]. The drug-resistant human leukemia K562 clonal cell line (i.e., designated K/VP.5) with acquired resistance (30-fold) to etoposide was cross-resistant to amsacrine, doxorubicin, mitoxantrone, and teniposide compared to parental K562 cells [96]. The multi-drug resistance observed was not mediated by the overexpression of ABCB1 (ATP binding cassette subfamily B member 1, a member of the superfamily of ATP-binding cassette [ABC] transporter) or due to mutations in the TOP2α gene in K/VP.5 cells [96,97]. Immunoblotting experiments utilizing a TOP2α antibody generated against amino acids 14–27 (i.e., N-terminal specific) indicated the presence of a novel 90 kDa isoform, TOP2α/90, along with the expected wild type TOP2α/170 [21,22]. Importantly, when compared to K562 cells, K/VP.5 TOP2α/170 protein levels were decreased ~90% and the expression of the TOP2α/90 isoform was increased ~three-fold [21,22].

Further experiments demonstrated that the C-terminal truncated TOP2α/90 isoform (786 aa and a calculated molecular weight of 90,076 Da) (Figure 7A–C) was the translation product of a TOP2α mRNA (2762 nt) that was generated from a cryptic PAS harbored in intron 19 (I19 IPA) (Figure 7B) [21,22]. As a result of I19 IPA, the TOP2α/90 mRNA shares only the first 19 exons with the TOP2α/170 transcript, is missing exons 20–35, and harbors a novel 3′-UTR (302 nt) (Figure 7B) [21,22]. Thus, when the TOP2α/90 mRNA transcript is translated, the resulting truncated TOP2α/90 isoform is missing 770 aa present in the C-terminus of TOP2α/170, which are replaced by 25 unique aa encoded by translation of the exon 19/intron 19 (E19/I19) “read-through” (Figure 5 and Figure 7) [21,22]. TOP2α/90 lacks the active site tyrosine (i.e., Tyr805, encoded by exon 20) (Figure 5C and Figure 7) and cannot form TOP2α–DNA covalent complexes or directly induce DNA breaks [1,2,3,4].

Due to I19 IPA, the TOP2α/90 isoform is also missing two dimerization domains (aa 1053–1069, 1121–1143) [98,99,100,101,102], and the NLSs (aa 1259–1296 and 1454–1497) [95,96] found in the TOP2α/170 isoform (Figure 5C and Figure 7) [21,22]. Since TOP2α subunit homodimerization and nuclear localization are necessary for enzymatic activity [1,2,3,4], the absence of these domains would be expected to prevent the formation of TOP2α/90:TOP2α/170 heterodimers and/or the localization of the TOP2α/90 isoform to the nucleus.

Unexpectedly, TOP2α/90 was detected and found primarily in the nucleus by immunofluorescence in intact cells and by immunoassays using nuclear and cytoplasmic extracts [22]. Additionally, coimmunoprecipitation experiments demonstrated that, even with the absence of canonical dimerization domains, the endogenous truncated TOP2α/90 isoform heterodimerized with the wildtype TOP2α/170 isoform [22]. This observation was consistent with several studies demonstrating that human N-terminal TOP2α ATPase domain fragments (aa 1–435; see Figure 5 and Figure 7), dimerize in vitro under the appropriate conditions [102,103,104,105]. At present, it is unclear how the TOP2α/90 isoform gains entry into the nucleus. Formation of TOP2α/90:TOP2α/170 heterodimers in the cytoplasm may allow TOP2α/170 to carry TOP2α/90 into the nucleus by a “piggy-back mechanism” [106]. Alternatively, if TOP2α/90:TOP2α/170 heterodimers form in the nucleus, this would suggest that the TOP2α/90 isoform is imported into the nucleus by an uncharacterized NLS sequence.

Since TOP2α/90 heterodimerizes with TOP2α/170, gains entry to the nucleus, yet lacks the ability to alter the topologic states of DNA molecules (i.e., no active site Tyr805), it was posited that the truncated TOP2α/90 isoform would exert dominant-negative effects on anticancer drug activity. Consistent with this hypothesis, forced expression of TOP2α/90 in parental K562 cells demonstrated that etoposide-mediated DNA strand breaks and cytotoxicity were suppressed [21,22]. Conversely, siRNA-mediated knockdown of TOP2α/90 in K/VP.5 cells enhanced etoposide-induced DNA damage [22]. Overall, these studies indicated that TOP2α/90 functioned as a resistance determinant [21,22].

Currently, it is not clear why the TOP2α I19 IPA event is favored in etoposide-resistant K/VP.5 cells. TOP2α sequencing in K/VP.5 cells revealed that no mutations were created in this gene during the generation (i.e., intermittent then continuous treatment with 0.5 μM etoposide) of this resistant clonal cell line [96,97]. Therefore, it is speculated that the observed increase TOP2α I19 IPA in K/VP.5 cells is likely due to aberrant expression/regulation of splicing/RNA cleavage/polyadenylation factors in this acquired drug-resistant cell line.

## 9. TOP2α/90 IPA in K/VP.5 Cells: Using CRISPR/Cas9/HDR to Circumvent Drug Resistance

Alternative splicing and intron retention may be regulated by the strength or relative weakness of the SS at the 5′ and/or 3′ ends of the intron, which can impede the spliceosome’s ability to recognize introns that should be spliced out [44]. IPA is frequently associated with weak 5′ SS, large introns, and strong PAS located ~100–1000 bp downstream from the 5′ SS of an intron [32,33,34]. It was hypothesized that strengthening a weak or suboptimal 5′ SS through mutagenesis would improve the complementarity between the U1 snRNA resulting in more efficient recruitment of the U1 snRNP complex (Figure 1A,B, Complex E) and subsequently increase the efficacy of splicing out a given intron (i.e., inhibit intron retention and/or IPA) [107,108,109,110,111,112,113]. For example, Yue and Ogawa [113] utilized CRISPR/Cas9 with HDR to introduce mutations at the 5′ SS of intron 7 of the short Xist (X specific transcripts [*Mus musculus*]) long noncoding RNA isoform to successfully improve splicing efficiency to regulate expression of the long form of Xist.

Given that TOP2α I19 IPA frequency is increased in K/VP.5 cells, the TOP2α E19/19 5′ SS (GAG//GTAAAC) was subjected to SS analysis (Splice Site Score Calculation; http://rulai.cshl.edu/new_alt_exon_db2/HTML/score.html; initially accessed 15 March 2020). The SS score for TOP2α E19/19 5′ SS was suboptimal, with a score of 6.1 (Figure 8A, blue box) out of a maximum score of 12.4 for the optimal consensus 5′ SS (CAG//GTAAGT). The three nucleotide differences which account for the differences in SS scores are bolded and underlined. Therefore, it was speculated that the suboptimal TOP2α E19/19 5′ SS influences I19 IPA [23]. It was further posited that by optimizing this SS by CRISPR/Cas9 with HDR in etoposide-resistant K/VP.5 cells, the U1 snRNP complex would be recruited more efficiently, TOP2α I19 IPA would be decreased, and sensitivity to etoposide would be restored [23].

Gene editing using CRISPR/Cas9 requires a guide RNA (gRNA) containing both a crRNA (CRISPR RNA) for DNA targeting and a tracrRNA (72-bp trans-activating CRISPRRNA) for nuclease activity [24,25,26,27,28]. To target the Cas9 nuclease to the DNA loci, crRNA must be complementary to targeted DNA (shown in red; Figure 8A). In addition, a sequence motif (NGG), the protospacer-adjacent motif (PAM, shown in green), must be present in the targeted locus (Figure 8A) [24,25,26]. The Cas9 nuclease introduces blunt-end double strand breaks (DSBs) three bases upstream of the PAM (Figure 8A, labeled in red with the Cas9 cut site denoted with a red arrow) [24,25,26]. Cas9-induced DSBs are predominantly repaired by the error-prone nonhomologous end joining (NHEJ), which results in nonspecific insertions/deletions (Indels) [114] often used for knocking out gene expression. However, exogenous custom templates can be utilized to repair Cas9-induced DSBs by HDR, thus allowing knock-in of specific mutations (Figure 8C) [25]. Importantly, gRNA targeting close to the intended mutation is required to obtain high editing efficiency by HDR [28,115].

Because the TOP2α E19/I19 5′ SS (GAG//GTAAAC) present in both K562 and K/VP.5 cells is suboptimal (Figure 8A, blue box), algorithm analyses were undertaken to determine the impact of specific gene edits on the 5′ SS scores [23]. It was determined that, by editing only the last two nucleotides of the TOP2α E19/I19 5′ SS (AC → GT, denoted in blue and underlined), the 5′ SS score was enhanced (from 6.1 → 11.6) (Figure 8B, blue box) [23]. An HDR repair template was synthesized to include the AC → GT alterations and a G → C modification to eliminate the PAM site (Figure 8A–C). This strategy was employed to avoid the recutting of already edited alleles upon subsequent rounds of transfection required to edit all three alleles of the TOP2α gene known to be present in both K562 cells [116,117] and the clonal K/VP.5 cells.

After etoposide-resistant K/VP.5 cells were co-transfected with gRNA, Cas9, and the HDR template, the cells were seeded at 0.8 cells per well (96 well plates) and single-colony wells were screened by genomic DNA qPCR to identify clonal cell lines with mutated TOP2α alleles [23]. After two rounds of transfection, a clonal cell line was identified, by qPCR and Sanger sequencing, that harbored three TOP2α CRISPR/Cas9/HDR edited alleles (designated K/VP.5/edit-3 cells) (Figure 9) [23]. Characterization of the K/VP.5/edit-3 cells by RNA-seq and immunoassays revealed decreased TOP2α I19 IPA (e.g., decreased TOP2α/90 mRNA/protein expression), improved splicing out of I19, and increased TOP2α/170 mRNA/protein expression (Figure 10A,B) [23]. Functional studies demonstrated that sensitivity to etoposide-induced DNA damage (results not shown; [23]) and etoposide-induced growth inhibition were restored in K/VP.5/edit-3 cells to levels comparable to those in parental K562 cells (Figure 10C) [23]. Additionally, complete circumvention of resistance was observed with other TOP2α interfacial inhibitors/poisons, including teniposide and daunorubicin in K/VP.5/edit-3 cells, whereas resistance was extensively but not completely reversed with mAMSA, mitoxantrone, and pixantrone [23].

These results indicated that CRISPR/Cas9/HDR gene editing of a suboptimal E19/I19 5′ SS in the TOP2α gene resulted in circumvention of acquired drug resistance to etoposide (Figure 10C) and other TOP2α-targeted drugs [23] in K/VP.5 cells by decreasing TOP2α I19 IPA. Consequently, the synthesis of the truncated TOP2α 90 kDa isoform (i.e., which is unable to form TOP2α–DNA covalent complexes or directly induce DNA breaks) decreased. Concomitantly, the expression full-length TOP2α/170 which was increased, restored sensitivity to TOP2α-targeted drugs.

## 10. Conclusions

Aberrant alternative splicing and polyadenylation can influence malignant transformation, development, progression, and cancer chemoresistance at the posttranscriptional level by altering mRNA stability, translational efficiency, nuclear export, mRNA localization, and by generating mRNAs which encode novel protein isoforms with divergent structures and functions [34,35,39,50,51,52,60,61,62,63]. Alternative splicing/polyadenylation seems to be the case with documented C-terminal TOP2α truncated isoforms that are encoded by mRNA transcripts generated by IPA and that play a role in mediating chemoresistance to TOP2α interfacial inhibitors/poisons [16,19,21,22,23]. Importantly, optimization of the weak TOP2α exon 19/intron 19 5′ SS in drug-resistant K/VP.5 cells by gene-editing decreased TOP2α I19 IPA, thereby diminishing TOP2α/90 expression, restoring TOP2α/170 levels, and circumventing drug resistance [23]. These results suggest that CRISPR/Cas9 with HDR methodology, in the future, may provide a novel therapeutic approach not only to strengthen SS to decrease IPA, a tactic that our laboratory has used successfully [23], but could also be used to edit and “correct” mutations within SS that disrupt normal splicing, introduce SS, and/or disrupt splicing regulatory *cis*-elements harbored in exons and introns often observed in inherited cancer genes [118,119].

To successfully treat human monogenic disorders utilizing CRISPR/Cas-mediated ex vivo or in vivo genome therapy, many hurdles need to be overcome [120]. These include optimizing CRISPR/Cas delivery systems (e.g., adeno-associated vectors [AAV] encoding Cas, gRNA, and an HDR donor template, or lipid nanoparticles [LNPs] to deliver these components), enhancing HDR efficiency, diminishing the immunogenicity of gene editing components, and increasing selectivity to attenuate off-target effects. As these challenges to therapeutic implementation are met, in vivo CRISPR gene editing may become commonplace in a clinical setting.

## Figures and Tables

**Figure 1 cancers-14-03148-f001:**
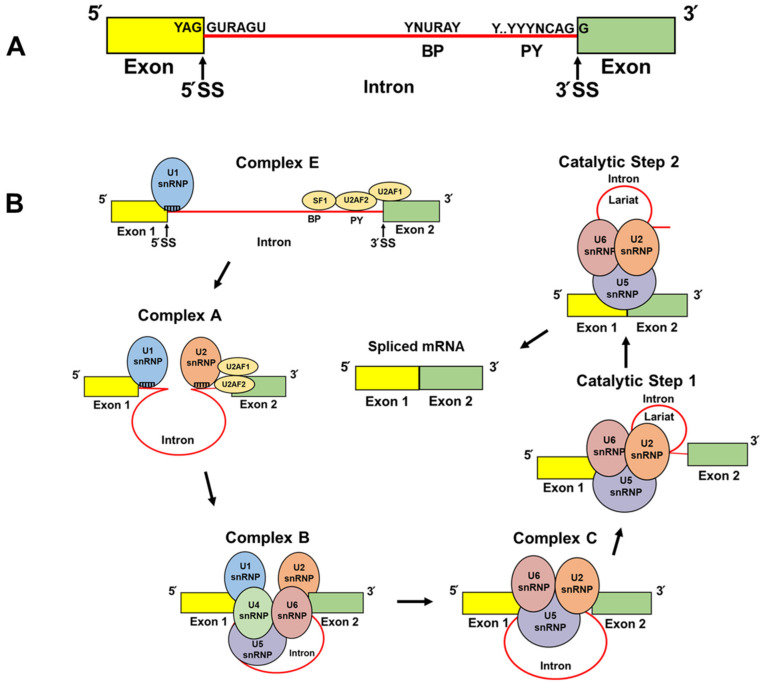
Schematic representations of core splice site sequence elements and spliceosomal assembly and action. (**A**) Two exons (yellow and green) separated by an intron (shown in red). The positions and consensus sequences of the 5′ splice site (5′ SS), branch point (BP), polypyrimidine tract (PY), and 3′ splice site (3′ SS) are denoted. R symbolizes nucleotides A and G. Y symbolizes nucleotides C and U. N represents any purine or pyrimidine nucleotide. (**B**) Depicted are the simplified stepwise interactions of the U2AF heterodimer and SF1 to the BP, PY, and 3′ SS and the subsequent recruitment of the spliceosomal small nuclear ribonucleoprotein (snRNP) particles (U1, U2, U4, U5, and U6) [36,37,38]. The names of the complexes, and the first and second catalytic steps, are labeled.

**Figure 2 cancers-14-03148-f002:**
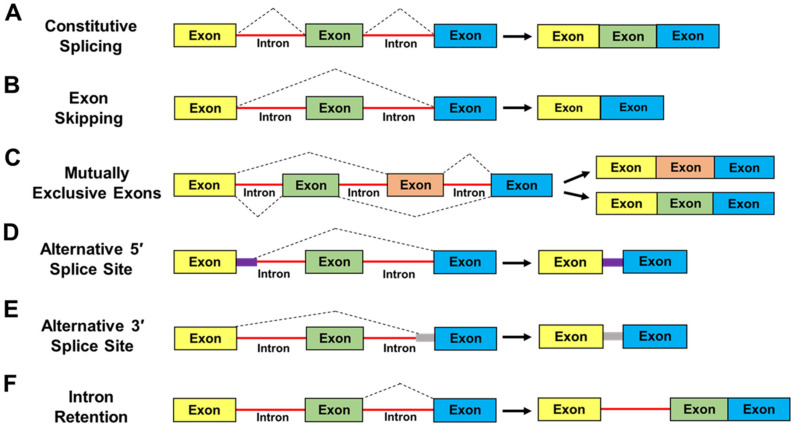
Schematic representation of classical mechanisms of alternative splicing. The figure displays different classes of alternative mRNA processing events and examples of alternatively spliced products [36,37,38,44]. (**A**) Constitutive Splicing; (**B**) Exon Skipping; (**C**) Mutually Exclusive Exons; (**D**) Alternative 5′ Splice site; (**E**) Alternative 3′ Splice site; (**F**) Intron Retention. Further information regarding each class of alternative splicing events is given in the text.

**Figure 3 cancers-14-03148-f003:**
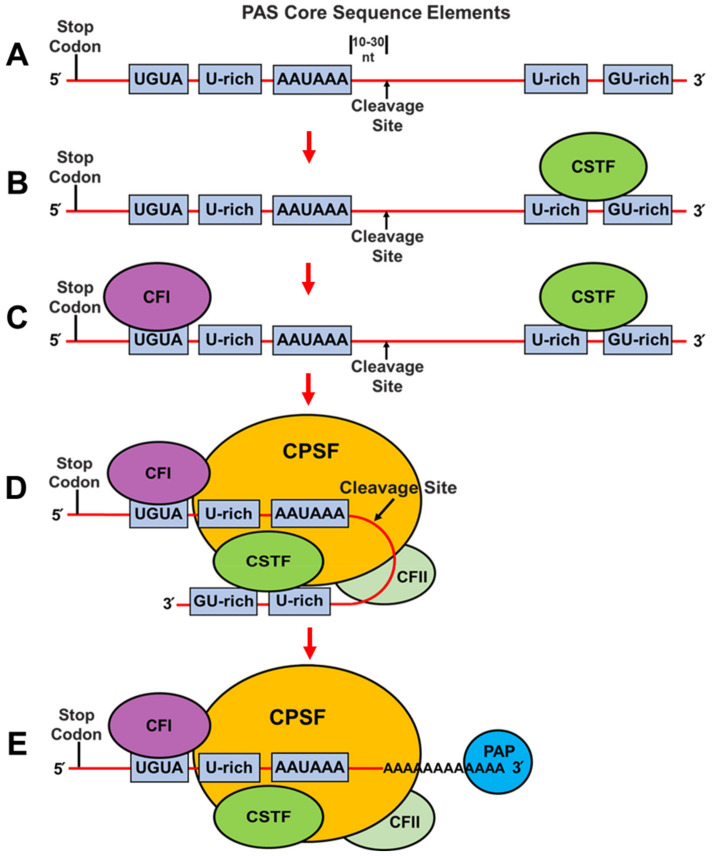
Schematic representation of core sequence elements and factors involved in cleavage and polyadenylation. (**A**) Cleavage and polyadenylation are regulated by *cis*-elements which include the AAUAAA hexamer, U-rich elements and UGUA elements upstream and U-rich and GU-rich elements located downstream from the hexamer, respectively. (**B**–**E**) Depicted are the stepwise interactions of the CSTF (cleavage and stimulation factor), CFI (cleavage factor I), CPSF (cleavage and polyadenylation specificity factor), CFII (cleavage factor II) complexes, and PAP (poly(A) polymerase) with specific *cis*-elements and with each other [32,33,34,35]. More details regarding the individual complexes and their specific interactions are given in the text.

**Figure 4 cancers-14-03148-f004:**
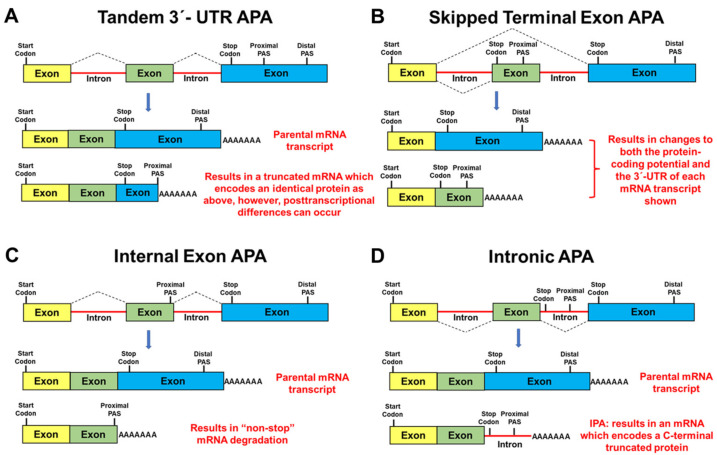
Schematic representation of mechanisms of alternative polyadenylation (APA). The figure displays different categories of alternative polyadenylation events [32,33,34,35]. (**A**) Tandem 3′-UTR APA; (**B**) Skipped Terminal Exon APA; (**C**) Internal Exon IPA; (**D**) Intronic APA. Further information regarding each class of alternative polyadenylation events is given in the text.

**Figure 5 cancers-14-03148-f005:**
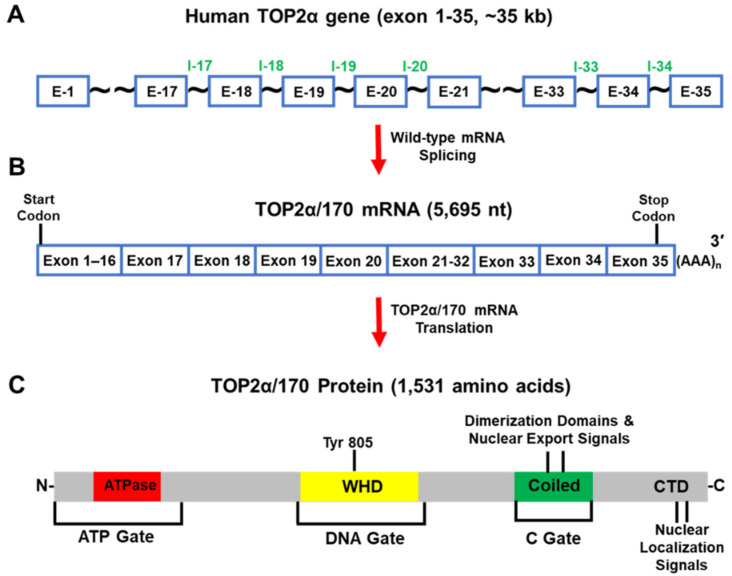
Schematic representation of the human TOP2α gene, TOP2α/170 mRNA, and TOP2α/170 isoform. (**A**) The human TOP2α gene comprises 35 exons and spans ~35 kb. (**B**) The well-characterized TOP2α/170 mRNA transcript is 5695 nucleotides long. (**C**) This mRNA transcript encodes a TOP2α isoform of 1531 aa, denoted TOP2α/170. The TOP2α/170 isoform harbors the ATP-gate/ATPase domain; the DNA-gate which includes the active site Tyr-805 and a winged-helix domain (WHD); the C-gate containing protomer dimerization sequences and nuclear export signals; the C-terminal domain (CTD) which contains the nuclear localization signals.

**Figure 6 cancers-14-03148-f006:**
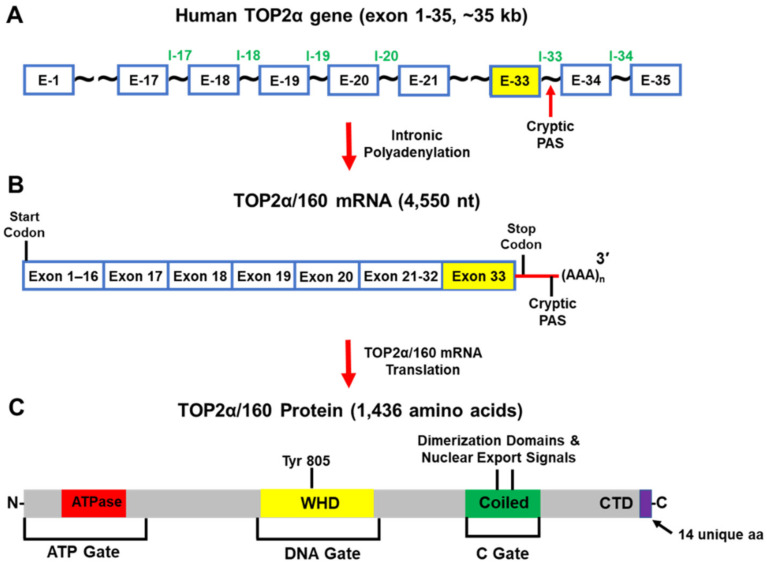
Schematic representation of the human TOP2α gene, TOP2α/160 mRNA, and TOP2α/160 isoform. (**A**) The human TOP2α gene comprises of 35 exons and spans ~35 kb. (**B**) When Intron 33 IPA occurs, the predominate TOP2α mRNA transcript is 4550 nucleotides long. (**C**) This mRNA encodes a TOP2α isoform of 1436 aa, denoted TOP2α/160. The truncated TOP2α/160 isoform is missing the nuclear localization signal (NLS 1454-1497) and is found predominantly in the cytoplasm. Therefore, the number of TOP2α/170 DNA covalent cleavage complexes are reduced and drug resistance occurs.

**Figure 7 cancers-14-03148-f007:**
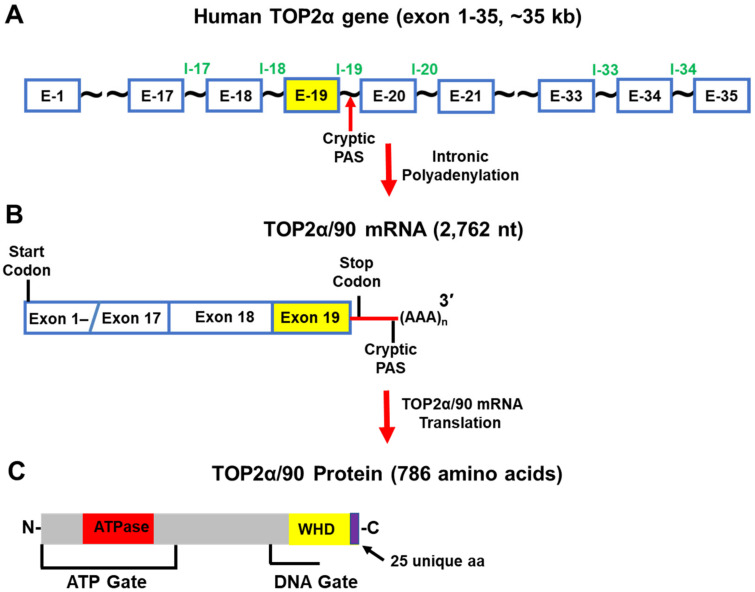
Schematic representation of the human TOP2α gene, TOP2α/90 mRNA, and TOP2α/90 protein. (**A**) The human TOP2α gene comprises 35 exons and spans ~35 kb. (**B**) When Intron 19 IPA occurs, the predominate TOP2α mRNA transcript is 2762 nucleotides long. (**C**) This mRNA transcript encodes a TOP2α isoform of 786 aa, denoted TOP2α/90. The truncated TOP2α isoform is missing the entire C-terminus (770 aa) present in TOP2α/170 and lacks the active site Tyr805. The TOP2α/90 isoform cannot form DNA covalent cleavage complexes and drug resistance occurs due to decreased drug-induced DNA damage, and cytotoxicity.

**Figure 8 cancers-14-03148-f008:**
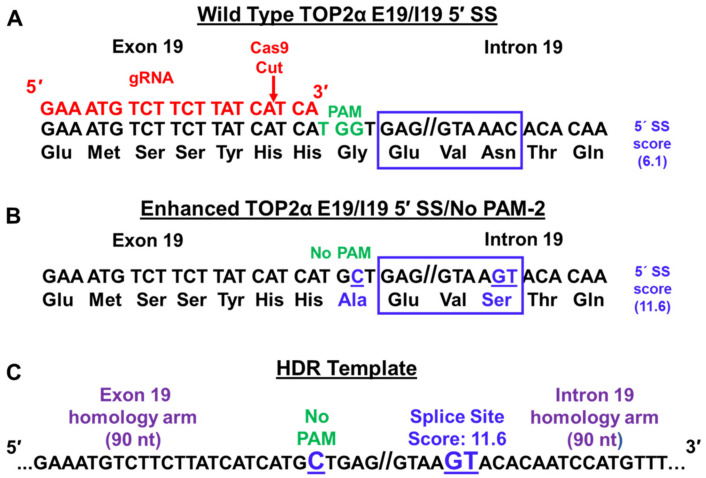
CRISPR/Cas9 and HDR strategy for editing TOP2α E19/I19 5′ SS and PAM site. (**A**) Shown is the TOP2α E19/I19 gene boundary sequence along with the E19/I19 5′ SS sequence (blue box) to be edited. The PAM site is denoted in green. The gRNA sequence is shown in red. The red arrow denotes where Cas9 generates a DSB. (**B**) Proposed changes to “enhance” the E19/I19 5′ SS (blue box underlined in blue) and to silence the PAM site (TGG → TGC, blue underlined) are indicated. The “improved” SS score as well as amino acid changes (blue) are denoted. (**C**) A symmetric 180-nucleotide repair template (HDR template) containing proposed changes (bolded, underlined in blue) in the TOP2α E19/I19 5′ SS and PAM sites is shown. Republished with permission of the American Society for Pharmacology and Experimental Therapeutics, from Hernandez et al. Mol Pharmacol. 99:226–241, 2021 [23].

**Figure 9 cancers-14-03148-f009:**
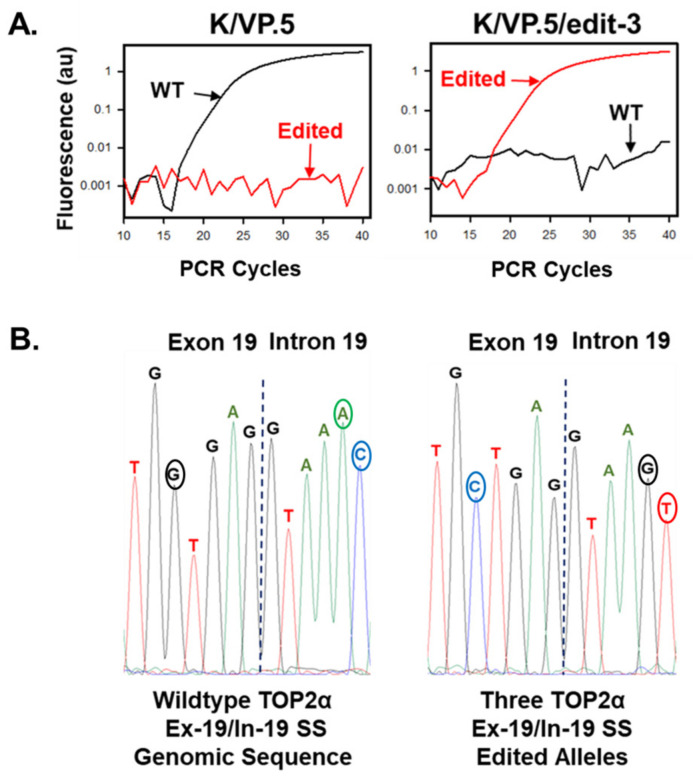
Selection and identification of a CRISPR/Cas9-edited TOP2α E19/I19 5′ SS clonal cell line. (**A**) qPCR results from K/VP.5 and K/VP.5/edit-3 cells using wild-type and edited specific E19/I19 boundary qPCR probes. (**B**) Genomic sequencing of the TOP2α E19/I19 boundary in K/VP.5 and K/P.5/edit-3 cells. Republished with permission of the American Society for Pharmacology and Experimental Therapeutics, from Hernandez et al. Mol Pharmacol. 99:226–241, 2021 [23].

**Figure 10 cancers-14-03148-f010:**
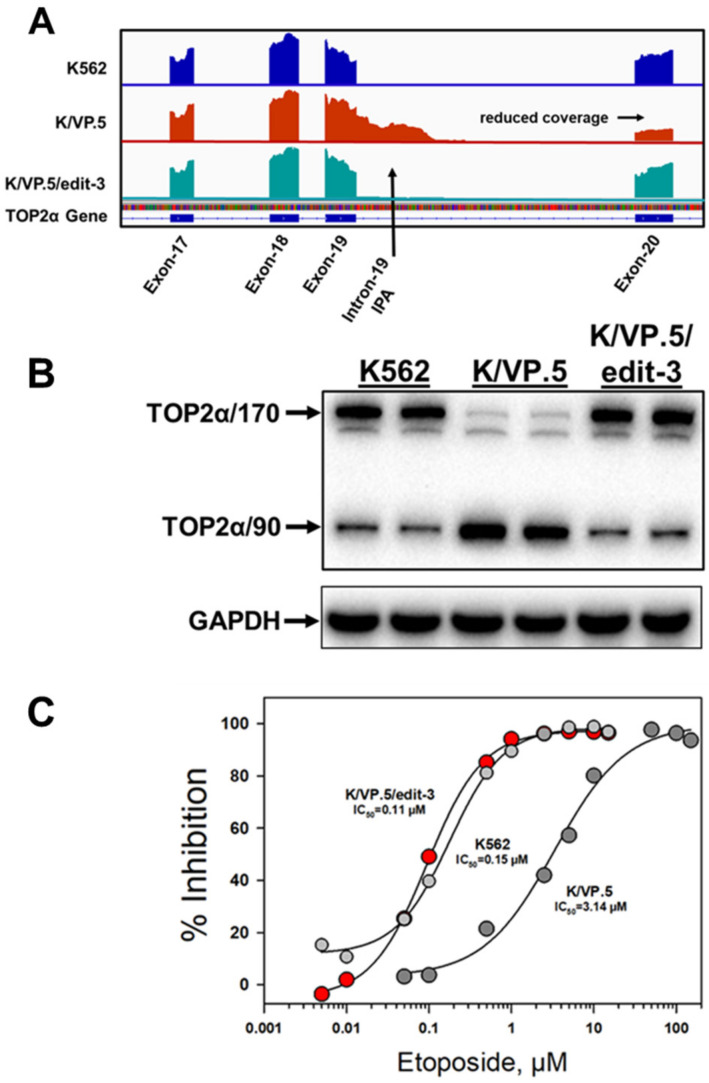
Effects of optimizing the TOP2α intron 19 5′ SS in K/VP.5 cells by CRISPR/Cas9/HDR editing. (**A**) RNA-seq genome coverage tracks for TOP2α showing the intron 19 IPA event in K/VP.5 cells and the restoration of intron removal in K/VP.5/edit-3 cells. Reduced coverage in exon 20 in K/VP.5 cells is consistent with fewer full-length TOP2α/170 reads while greater exon 20 coverage in K/VP.5/edit-3 cells is consistent with intron 19 removal and more full-length mature TOP2α/170 mRNA. (**B**) Immunoassay (N-terminal specific TOP2α antibody; GAPDH antibody) using K562, K/VP.5, and KVP.5/edit-3 cellular lysates. (**C**) Forty-eight-hour growth inhibitory effects of etoposide in K562, K/VP.5, and K/VP.5/edit-3 cells. Republished with permission of the American Society for Pharmacology and Experimental Therapeutics, from Hernandez et al. Mol Pharmacol. 99:226–241, 2021 [23].

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
