# Peer review of "Intronic Polyadenylation in Acquired Cancer Drug Resistance Circumvented by Utilizing CRISPR/Cas9 with Homology-Directed Repair: The Tale of Human DNA Topoisomerase IIα"

_cancers, 2022, doi:10.3390/cancers14133148_

Round 1

Reviewer 1 Report

The article by Elton et al. is well presented, placed in a highly relevant context, cancer-linked cellular adaptation to treatment by DNA topoisomerase II poisons. Some of the resistance to TOP2α/170 inhibitory mechanisms involve specific alterations of the corresponding transcripts, resulting in production, by intronic polyadenylation in intron 33 or 9, respectively, of truncated, dominant-negatively acting TOP2α/160 and TOP2α/90 isoforms. Using CRISPR/Cas9/HDR in a TOP2α/90 expression context, restoration of sensitivity to TOP2α-targeted drugs was achieved.

The contents of the article are mostly relevant. However, as presented, it reads as a mixture of review/general information (spliceosome cycle, types of splicing events, but then fails to describe similarly the Nonsense Mediated mRNA Decay mechanism) and of a focused content (mostly published before, see below), which is the important part, taking the case of TOP2a alterations as the article main aim. I would thus suggest to remove figures 1, 2 and shorten the corresponding paragraphs, as the literature is filled with equivalent figures. In addition, figures 8, 9 and 10, which have also been published before (and reproduced here with permission), should also be removed.

Author Response

We thank the reviewer for positive comments indicating that our article is “well-presented” and is “placed in a highly relevant context”. 

In response to the reviewer’s concern that there is a lack of discussion of nonsense-mediated decay (NMD), we have included information concerning NMD as a sequelae of intron retention (lines 130-132).

We prefer to retain Figure 1 since the simplified depiction (and textual review) of splicing mechanism(s) is of import in understanding the basis of and effectors for intronic polyadenylation.

We similarly prefer to retain Figure 2 since classical alternative splicing mechanisms are presented as a contrast to subsequent review of processes related to alternative polyadenylation (Figure 4).

We respectfully disagree with the reviewer’s recommendation to remove Figures 8-10.  These figures are key for readership understanding of the tale of TOP2α regarding an implemented strategy to identify and characterize gene-edited clones for successful circumvention of drug-resistance.

Reviewer 2 Report

This paper clearly addressed the mechanism of homolog-directed repair for overcoming the drug resistance of topoisomerase IIA inhibitor in leukemia cell lines. It provided an insightful approach to chemotherapeutic treatment. There are a few things that need to be addressed. TOP2a inhibitors have been used in a variety of cancer treatments, why focus on leukemia cells in particular? Besides etoposide, will the drug resistance could be circumvented under other TOP2a inhibitors treatment? 

Author Response

We thank this reviewer for the overall positive comments especially the indication that our article, “provided and insightful approach to chemotherapeutic treatment”.

We have focused our article on intronic polyadenylation in acquired drug resistance telling the tale of TOP2α in leukemia cells since, to date, there have been a dearth of investigations into this mechanism in other cancer types. Future studies by our group (and hopefully others) will evaluate this mechanism of resistance and its circumvention.

The reviewer has asked about circumvention of resistance to other TOP2α inhibitor besides etoposide.  In a recent publication (Reference 23),  complete or nearly complete circumvention of resistance was demonstrated to a number of TOP2α-targeted inhibitors including daunorubicin, teniposide, mAMSA, mitoxantrone, and pixantrone.  We have now included this information in the revised manuscript (lines 465-468).

Reviewer 3 Report

Dear Authors,

the review entitled “Intronic  Polyadenylation in Acquired Cancer Drug Resistance Circumvented by Utilizing CRISPR/Cas9 with Homology-Directed Repair: The Tale of Human DNA Topoisomerase IIα” provides an overview of the different alternative splicing mechanisms, focusing, in particular, on aberrant alternative splicing and intronic polyadenylation (IPA) as responsible in some cases of acquired resistance to DNA topoisomerase IIα (TOP2α) inhibitors/poisons, and shows how the use of CRISPR/Cas9 with HDR may overcome acquired resistance to this class of drugs in an in vitro system.

 I find the manuscript interesting and well written, also concerning the topic of chemotherapy resistance which affects the major part of cancer patients, offering a potential, even if still far from the” bedside”, therapeutic approach to overcome this oncological issue. I suggest stressing better how the use of CRISPR/Cas9 technique may be translated to the clinical practice.

Overall, I would recommend the publication in Cancers if the authors can kindly address my comment and the following minor comments:

- Line 51: “and solid tumors,” remove “,” after tumors

-Line 77: in A stepwise fashion

- Please, write in italic the name of the genes (i.e. TOP2α at line 261)

- Line 285: Remove the “.” after “strands of DNA.”

- Line 367: missing “  after piggy-back mechanism

Author Response

We thank the reviewer for the comment that our manuscript was “interesting and well written”. 

The reviewer requests additional indications stressing how CRISPR/Cas9 gene-editing may be translated clinical applicability.  In response, we have included an additional final paragraph indicating the potential (and challenges) of clinical translation using CRISPR/Cas 9 (lines 511-518).

All minor corrections pointed out by the review as minor comments have been made and indicated in track changes (highlighted in yellow in the revised manuscript; lines 51,77,269, 293, 377).